# Glial Populations in the Human Brain Following Ischemic Injury

**DOI:** 10.3390/biomedicines11092332

**Published:** 2023-08-22

**Authors:** Victoria Mihailova, Irina I. Stoyanova, Anton B. Tonchev

**Affiliations:** Department of Anatomy and Cell Biology, Faculty of Medicine, Medical University Varna, 9000 Varna, Bulgaria; stoyanovai@yahoo.co.uk (I.I.S.); anton.tonchev@mu-varna.bg (A.B.T.)

**Keywords:** glial cells, astrocytes, oligodendrocytes, NG2 glia, microglia, human brain, ischemic brain injury, transcription factors

## Abstract

There is a growing interest in glial cells in the central nervous system due to their important role in maintaining brain homeostasis under physiological conditions and after injury. A significant amount of evidence has been accumulated regarding their capacity to exert either pro-inflammatory or anti-inflammatory effects under different pathological conditions. In combination with their proliferative potential, they contribute not only to the limitation of brain damage and tissue remodeling but also to neuronal repair and synaptic recovery. Moreover, reactive glial cells can modulate the processes of neurogenesis, neuronal differentiation, and migration of neurons in the existing neural circuits in the adult brain. By discovering precise signals within specific niches, the regulation of sequential processes in adult neurogenesis holds the potential to unlock strategies that can stimulate the generation of functional neurons, whether in response to injury or as a means of addressing degenerative neurological conditions. Cerebral ischemic stroke, a condition falling within the realm of acute vascular disorders affecting the circulation in the brain, stands as a prominent global cause of disability and mortality. Extensive investigations into glial plasticity and their intricate interactions with other cells in the central nervous system have predominantly relied on studies conducted on experimental animals, including rodents and primates. However, valuable insights have also been gleaned from in vivo studies involving poststroke patients, utilizing highly specialized imaging techniques. Following the attempts to map brain cells, the role of various transcription factors in modulating gene expression in response to cerebral ischemia is gaining increasing popularity. Although the results obtained thus far remain incomplete and occasionally ambiguous, they serve as a solid foundation for the development of strategies aimed at influencing the recovery process after ischemic brain injury.

## 1. Origin of Macroglial Cells

Тhe brain of adult mammals has long been considered a “static” organ that does not change structurally throughout life. Pioneers in neuropathology believed that brain tissue could only undergo necrosis because neither neurons nor glia had regenerative potential [1]. Several scientific discoveries in recent decades have pointed to the hypothesis that mature brain tissue exhibits plasticity, and its cellular composition undergoes dynamic quantitative changes both under physiological conditions and in various pathological states. It is now known that new neurons and glial cells are generated in different regions of the adult brain, originating from a variety of precursor cells, and this process continues throughout an individual’s lifetime [2,3].

Two regions have been identified in the brains of mammals where so-called neuronal stem cells (NSCs) can be found [3]. These are specialized progenitor cells capable of both self-renewal and differentiation into multiple cell types [3]. One of these regions is the hippocampal gyrus dentatus (DG), where NSCs form a layer located beneath the granule cell layer, hence referred to as the subgranular zone (SGZ) [2]. NSCs can also be found in the ventricular-subventricular zone (VZ-SVZ), which is a thin layer of brain tissue lining the wall of the lateral ventricle in the telencephalon [4]. These two areas are defined as “neurogenic” niches in the brains of mammals, including humans, as it has been demonstrated that new neurons are generated from neural stem cells during the postnatal period [3]. NSCs share some morphological and molecular characteristics with specialized glial cells, such as radial glia (RGCs) or astrocytes in the adult brain [5,6]. For example, all these cell types express glial fibrillary acidic protein (GFAP) [7].

The fact that these cell lineages share similar markers makes studying the origin of astrocytes a challenging task [6,8]. At the same time, it has been shown that several transcription factors modulate astrocytic differentiation in the early stages of embryonic development and are also expressed in NSCs and other progenitors in the germinal zone of the embryonic and adult brain [9,10,11,12]. However, there are two main hypotheses regarding the origin of astrocytes. The first hypothesis suggests that astrocytic precursor cells are formed from immature progenitors such as NSCs, which migrate from the germinal zone to the brain parenchyma and differentiate into mature astrocytes at a later stage [13]. The second possible origin is through the transformation of RGCs into astrocytes. After fulfilling their primary function, the processes of radial glial cells retract, and they transform into freely dispersed cells in extensive areas of the central nervous system (CNS). It is believed that these cells are the ones that transform into astrocytes during the postnatal period since they have identical morphology and express identical markers. This scenario also supports the idea that NSCs and mature astrocytes originate from RGCs and therefore share common characteristics [8]. It is still unclear whether these two pathways of astrocytic differentiation coexist or if one of them predominates. What is certain is that NSCs, both from the hippocampal DG and the VZ-SVZ, generate a small number of new astrocytes in the mature brain [13,14,15,16,17].

Oligodendrocytes (OLs) are myelinating glial cells in the central nervous system (CNS). The precursor cells of oligodendrocytes are called oligodendrocyte progenitor cells (OPCs). In the early postnatal period, OPCs differentiate into pre-myelinating immature oligodendrocytes (pre-OLs), which gradually mature into myelin-expressing Ols [18]. A small number of OPCs are generated in the VZ-SVZ from NSCs in the fully mature brain, both in physiological and pathological conditions [19,20]. A significant portion of OPCs remains undifferentiated and distributed in various regions of the CNS in adult individuals [21,22]. A percentage of these de novo-generated cells in the adult brain or preexisting OPCs differentiate into mature OLs replacing the degenerated OLs under physiological or pathological conditions in the brains of rodents and humans [19,23]. In the past, it was believed that OPCs could only differentiate into OLs, but current in vivo studies have shown that, although limited in number, OPCs can differentiate into astrocytes and/or neurons in specific brain regions [24,25,26].

The question of whether NSCs are multipotent cells capable of differentiating into various cell lineages or whether there are bipotent NSCs with neuronal/astrocytic differentiation and monopotent NSCs that solely differentiate into one cell type remains controversial [27,28]. Recent studies suggest that some NSCs simultaneously produce neurons and astrocytes, while others may only generate one of the two cell lineages over a one-year period. During the same period, NSCs generating oligodendrocytes are not detected [14]. Thus, the debate continues as to whether NSCs are multipotent cells that choose different paths of differentiation or whether bipotent NSCs with neuronal/astrocytic differentiation and monopotent NSCs that exclusively differentiate into one cell type coexist.

In the adult brain, the presence of a specific glial subpopulation called NG2+ cells has been identified [29,30]. Initially, these cells were considered oligodendrocytic precursor cells. However, in the past few decades, it has been recognized that they represent a third type of macroglial cells in addition to astrocytes and oligodendrocytes. NG2 glia have been found in both the developing and adult brains of experimental animals and humans. They constitute 5–10% of the entire glial cell population and are distributed in both gray and white matter of the brain. NG2 glial cells retain their proliferative and differentiating capacity in mature brain tissue, thus representing the largest proliferating cell population outside of the neurogenic niches in the CNS. Studies involving the mapping of brain cell populations have revealed that NG2 glia in the white matter can differentiate into myelinating OLs, while NG2 glia in the gray matter do not differentiate into OLs. Other experiments suggest that this cell population may give rise to a subpopulation of astrocytes in the ventrolateral prosencephalon during embryonic development. The exact origin and role of NG2 glial cells in the CNS are still subjects of investigation [29,30].

## 2. Origin and Role of Other Cell Types in Postnatal Brain Development

Ependymal cells represent a monolayer that covers the surface of the lateral brain ventricles, thereby isolating the brain parenchyma from the ventricular system [31]. Some authors categorize ependymal cells as a separate cell category in the CNS, alongside neurons and macroglia. It is believed that they originate from primitive neuroepithelial cells distinct from RGCs. Therefore, ependymal cells are clearly distinguished from astrocytes in the CNS [32]. According to other authors, they actually arise from RGCs and are generated during embryonic development and the early postnatal period [33]. The diversity of ependymal cells is extensive, and there exist various classifications based on their morphology, localization, molecular profile, and functions [34]. What sparks curiosity and probably has relevance to their functioning not only in physiological but also in pathological conditions is that postnatal NSCs are interspersed among ependymal cells, with their cell bodies and/or apical processes found in both the ependymal and subependymal layers. This arrangement forms unique complex structures with the classical ependymal cells, observed exclusively near the surface of the lateral ventricles. Similar cytoarchitecture is not found in non-neurogenic areas of the CNS, such as the central canal of the spinal cord [35,36]. Whether this specific cellular organization has functional significance remains unclear.

The choroid plexus consists of capillaries covered by a single layer of ependymal cells from the brain ventricles. It is responsible for the production and reabsorption of cerebrospinal fluid [37]. The origin of the cells in the choroid plexus is analogous to that of ependymal cells, specifically from RGCs [38]. Recent studies indicate that the choroid plexus regulates the proliferation of NSCs in the VZ-SVZ of the adult brain [39,40].

The blood-brain barrier (BBB) is composed of vascular endothelial cells, pericytes, and astrocytes. There is evidence of interesting communication between the brain’s vascular network and various types of cells, including NSCs. It is believed that blood vessels can send different signals to NSCs and thereby control their proliferation and differentiation. Additionally, there is evidence that blood vessels regulate different stages of neuron and glial development. An interesting fact is that in the SVZ of the adult brain, NSCs interact with a dense network of specialized blood vessels located there [41,42].

According to current understanding, microglia originate from myeloid precursor cells, and their colonization of the brain tissue occurs during different stages of embryonic development. The majority of microglial cells migrate into the CNS during the postnatal period, following the formation of the BBB. It has been observed that microglia regulate the proliferation and differentiation of NSCs and OPCs by either stimulating or inhibiting them depending on the specific conditions. This way, microglia engage in direct interactions with cell populations in both the DG and the SVZ of the adult brain [43,44,45,46].

## 3. Astrocytes

Astrocytes were initially proposed as the “connective tissue” of the brain, composed of various cellular elements, by Rudolf Virchow in 1858 [47]. In 1895, Lenhossek first used the term “astrocytes” based on their morphology, characterized by multiple processes that give them a star-like appearance [48]. They communicate with each other through gap junctions located at the periphery of their processes. They also possess thin peripheral branches known as perisynaptic astrocytic processes that establish structural and functional relationships with synapses [49,50].

### 3.1. Morphological and Functional Heterogeneity

As described earlier, astrocytes likely originate from different cell types during embryonic and postnatal development, which contributes to their heterogeneity and potential variations based on their origins. The number of astrocytes varies in different brain regions, possibly reflecting their adaptation to the surrounding tissue conditions. In the 19th century, the presence of two main types of astrocytes with distinct localization was identified: protoplasmic astrocytes in the gray matter and fibrous astrocytes in the white matter. Protoplasmic astrocytes are evenly distributed and contact neurons, the pial surface of the brain, and the surface of brain capillaries, forming perivascular “sleeves” around blood vessels. Fibrous astrocytes are oriented parallel to the course of tracts in the white matter and have smaller cell bodies and processes that are not as numerous as those of protoplasmic astrocytes but are significantly longer. They make contact with myelinated nerve fibers in the regions of Ranvier’s nodes through their “perinodal” processes [51,52].

The contacts between fibrous astrocytes in the white matter are not as numerous as those between protoplasmic astrocytes in the gray matter. It is even possible to find individual unconnected fibrous astrocytes [53].

Other types of astrocytes include (1) velate astrocytes surrounding granule neurons in the cerebellum and olfactory bulb; (2) interlaminar astrocytes and astrocytes with varicose processes, which are specific to the cortical regions of primates, including humans; (3) tanycytes found in the third brain ventricle and the floor of the fourth ventricle; and (4) pituicytes, specialized astroglial cells in the hypothalamus. Perivascular and marginal astrocytes are different subtypes that participate in the formation of the BBB. Ependymal cells, the cells of the choroid plexus, and the retinal pigment epithelial cells, respectively, line the brain ventricles and the space beneath the retina. Specialized types of astroglia include Bergmann glia in the cerebellum and Müller glia in the retina [32,54]. The different astrocyte subtypes are presented in Figure 1. The density and morphology of astrocytes in different brain regions are determined by the cytoarchitecture and specific functional requirements of those regions [52].

The functions of astrocytes in the CNS are diverse. The connection between neurons and blood vessels is mediated by the terminal endings of astrocytes, which come into contact with both synapses and neuronal membranes on one side and the walls of blood vessels on the other [52]. This allows astrocytes to regulate blood pressure according to the energy and metabolic needs of synapses [55,56]. Additionally, astrocytes can alter the permeability of the vascular wall by releasing vasoactive substances and cytokines that control the water and ion balance in the brain tissue. Astrocytes also control K+ homeostasis and serve as a reservoir of glucose in the CNS [51]. Their heterogeneity is determined by the presence or absence of ion channels on their surface [57,58].

Interestingly, astrocytes possess excitability of the cell membrane, which is influenced by changes in calcium balance between the intracellular and extracellular space. They also have the ability to coordinate the propagation of membrane oscillations among themselves, known as calcium wave propagation. Both types of activity are resistant to blockers of electric potentials that can be secreted by neurons [59]. Intercellular calcium waves propagate differently in gray and white matter: in the cerebral cortex, gap junctions between astrocytes facilitate the spread of calcium waves, while in the corpus callosum, where there are fewer gap junctions between astrocytes, ATP is involved in the propagation of calcium waves [60]. However, in the neocortex, intact gap junctions between astrocytes are required for the rapid propagation of calcium waves [53].

Astrocytes are involved in the formation, maturation, maintenance, and control of functional synapses. They are sensitive to the metabolic demands of neurons and modulate neurotransmission by releasing and accumulating neurotransmitters, such as glutamate, as well as secreting purines (ATP, adenosine, guanine), GABA, and D-serine [61,62]. Astrocytes also exhibit heterogeneity in the expression of various membrane proteins (receptors) and transporter proteins associated with these functions [48].

### 3.2. Immunohistochemical Profile

Cell markers are used to study the morphology and function of cells under both physiological conditions and cellular damage. Numerous proteins specific to a particular cell type have been identified. Cell-specific antibodies have been developed based on this information for the identification of individual cell types. However, the results should be interpreted with caution. An increased or decreased number of positive cells can be observed following injury, as well as a change in the intensity of expression of the respective marker. At the same time, cell markers are suitable for detailed study of the morphology of individual cell types, especially in neural tissue, here they exhibit a pronounced diversity of their structure [63,64,65,66,67,68,69]. Figure 2 displays cellular markers used for the morphological categorization of astrocytes.

Proteins specific to astrocytes include vimentin, desmin, synemin, and glial fibrillary acidic protein (GFAP). They perform a mechanical function and are responsible for the structural integrity of astrocytes. However, the degree of their expression varies. Astrocytic intermediate filaments are composed of GFAP, making GFAP an established marker for studying their morphology over the years. It is expressed both in the cytoplasm and in astrocytic processes [64,66,68]. Larger cell processes are well visualized, while the use of an electron microscope is required for finer processes. However, it should be noted that not all astrocytic processes express it. Furthermore, in one of the astrocyte subpopulations, they are completely negative for GFAP [66]. Despite the aforementioned, this marker demonstrates high specificity toward the astrocytic glial population [64].

Another widely used marker for studying astrocytes is the cytoplasmic calcium-binding protein S100β. It is a secretory protein present in a higher percentage of astrocytes compared with GFAP, but it is also expressed in other cell types, including a small number of neurons and OLs [68,69]. One of the main functions of astrocytes is to take up excess glutamate from the extracellular space of the brain tissue and metabolize it into glutamine. This way, they regulate the excitability of neurons and synaptic connections. Excitatory amino acid transporter 1 (EAAT1) and 2 (EAAT2) are glutamate transport proteins located in astrocytes that prevent the accumulation of glutamate in the synaptic space. Therefore, anti-EAAT1/EAAT2 antibodies are used to label astrocytes in the CNS [67].

Astrocytes metabolize glutamate through the enzyme glutamine synthetase (GS). Changes in GS expression can be observed, which correlate with the amount of the enzyme in their cytoplasm and reflect changes in cellular functional activity [70]. Another antigenic marker with high specificity for astrocytes is aldehyde dehydrogenase family 1 member L1 (Aldh1L1), which participates in GABA metabolism [71].

Aquaporin-1 (AQP1) and Aquaporin-4 (AQP4) are transmembrane proteins that regulate water permeability across the cell membrane and maintain water balance and homeostasis in the brain tissue. AQP4 is expressed in the terminal segments of astrocytic processes, making it a suitable marker for their study as well as for the contacts that astrocytes form with other cells in the CNS [72].

The molecular heterogeneity of the astrocyte population can also be represented by the expression of CD44, a transmembrane protein and receptor for the extracellular matrix. Through CD44, morphological characteristics are revealed that cannot be described by GFAP alone. It has been observed that in humans, there is a group of astrocytes with long unbranched processes that do not have an equivalent in the rodent CNS. They differ from protoplasmic astrocytes in the gray matter specifically in the expression of CD44. Fibrous astrocytes in the white matter also express CD44. In the human neocortex, CD44+ astrocytes are most abundant in layer I, with fewer cells in the deeper layers of the cortex and in the white matter at the border with the cortex. Some of these cells in layers V–VI have varicose processes, while a small number of them in the middle layers do not have long processes. CD44+ astrocytes mainly contact larger blood vessels, but there are also those whose processes terminate among neuropil. CD44+ astrocytes have been found in various regions of the hippocampus, including the DG area. Based on their molecular profile, these cells differ from protoplasmic astrocytes, resembling the phenotype of fibrous astrocytes [73].

Based on everything presented so far, it can be summarized that the diversity in astrocyte morphology is significant and is accompanied by the presence of various combinations of cell markers. At the same time, transcriptomic analysis of individual cell populations can be an excellent tool for deciphering this heterogeneity [74].

### 3.3. Genetic Analysis

Zeisel et al. in 2015, through cellular sequencing, demonstrated the presence of two molecularly distinct subtypes of astrocytes in the mouse brain: type 1, located in the superficial layer I of the neocortex, which participates in the formation of the glial limiting membrane, and type 2, evenly distributed in the deeper layers of the cortex, with smaller cell body sizes and fewer cellular processes [74].

Subsequently, Morel et al. performed molecular profiling of astrocytes from six different brain regions and identified pronounced heterogeneity. Furthermore, they discovered that subcortical astrocytes express the gene secreted protein acidic and rich in cysteine (SPARC). This is associated with their characteristic function in this area, which involves promoting neurite growth and modulating synaptic contacts [75]. In parallel, Chai et al. (2017) conducted a comparative analysis of astrocyte functions in the hypothalamus and striatum, and the results obtained were similar, indicating that functional heterogeneity exhibits regional characteristics [76].

In 2017, Lin et al. developed a specialized technique to explore astrocytic diversity across multiple brain regions. It allowed for the differentiation of cell populations during embryonic brain development, but it was not applicable to mature brain tissue. Moreover, the technique did not enable morphological analysis or anatomical evaluation of the examined cell lines. Nonetheless, the researchers succeeded in subdividing the Aldh1l1-GFP astrocyte population into five distinct subpopulations, each present in five different CNS regions. Transcriptomic analysis of these five subpopulations revealed that each had a unique molecular profile. Region-specific molecular characteristics were also identified, supporting the idea of cell specialization based on their respective regions. Additionally, it was found that these subpopulations expressed genes responsible for synapse formation. Experiments with cell cultures of neurons and astrocytes demonstrated that the individual astrocyte subgroups contribute to synaptogenesis in different ways. Therefore, these astrocyte subgroups are not only molecularly but also functionally heterogeneous [77].

In an experimental study on wt mice using cellular sequencing (isolation of cells from specific brain regions for RNA analysis and profiling/isolation of transcribed microRNAs from astrocytes), in 2020, five subtypes of astrocytes with specific functions in different brain regions were identified. Cluster analysis confirmed the results obtained and the regional astrocytic heterogeneity by comparing them in the neocortex and hippocampus. The same analysis revealed the presence of three additional subtypes of mature astrocytes. The gene expression profile of this cell population is characteristic and clearly delineates the spatial distribution of individual subtypes. Furthermore, a fine gradient in gene expression is observed between different areas of the CNS. The diverse regional distribution and differences in gene expression suggest distinct and specific functions for astrocytic subtypes [78].

Subsequent research in 2021, again based on cellular sequencing in experimental animals, demonstrated that astrocytes are a heterogeneous population in terms of their genetic profile. They exhibit a characteristic regional distribution in the CNS and undergo dynamic changes during their development, which is reflected in their specific genetic profile at a certain stage of development. This suggests that these cells perform diverse functions during different periods of their existence [79].

### 3.4. Interspecies Heterogeneity

The morphological and functional heterogeneity of astrocytes was initially studied in rodents. Additional investigations in human and nonhuman primates have revealed certain characteristic interspecies differences. Specifically, astrocytes in the cortical gray matter of primates and humans are larger, more complex, and distinctly pleomorphic compared with their counterparts in rodents. There are also two separate types of astrocytes observed in humans and primates: interlaminar astrocytes (ILA) and varicose process astrocytes [80,81,82,83].

ILAs are found in Layer I of the cerebral cortex, with short processes extending to the pial surface and participating in the formation of the glial limiting membrane. They possess one or two longer processes (up to 1 mm) reaching Layers I–IV of the cerebral cortex, exhibiting a distinctive “bent” course and terminating on blood vessels or within neuropil. While ILAs in humans have a spherical body, those in primates have an elongated body and are less numerous [52]. Quantitative analysis of the number and morphological complexity of ILAs indicates that these two parameters increase during the postembryonic period. From birth to mature adulthood, the number of ILAs doubles in humans and quadruples in chimpanzees. In all primate species, the number of primary ILA processes does not change during development or at later stages, but the total length of all processes is the greatest in adults. It has been proposed that ILAs in the primate and human brain are generated during the embryonic period, most likely derived from RGCs, proliferate before birth, and exhibit some similar characteristics to rudimentary ILAs in rodents [80]. The exact role of this cell subtype is not yet clear. It is hypothesized that their long processes aid in the propagation of calcium waves in humans, thus increasing their speed, and that they play a role in maintaining the architecture and intercellular contacts in the cerebral cortex, as well as in the integrity of the blood-brain barrier and others [52,84].

In Layers V–VI of the cerebral cortex in humans and chimpanzees, there are GFAP+ astrocytes with varicose processes. These astrocytes possess one to five long processes (up to 1 mm) with a straighter course compared with protoplasmic astrocytes, and they terminate on blood vessels or within neuropil. Human astrocytes with varicose processes are larger and more complex compared with their counterparts in chimpanzees [80,82].

In the human cerebral cortex, protoplasmic astrocytes are larger and have longer processes, exhibiting a tenfold increase in the number of GFAP-positive processes compared with other species. Additionally, there is an elevated intensity of GFAP expression in their terminal segments. This suggests their involvement in a greater number of synaptic contacts. Protoplasmic astrocytes in humans, similar to those in rodents, are also organized into domains. Each astrocyte occupies a specific anatomical space and is partially overlapped by neighboring cells. Thus, the processes of only one astrocyte come into contact with the bodies of neurons, synapses, and blood vessels within one domain. Protoplasmic astrocytes in humans exhibit greater overlap due to their larger size. In rodents, one domain of astrocytes surrounds 20,000 to 120,000 synapses. Meanwhile, in humans, astrocytes within one domain can encompass between 270,000 to 2 million synapses. Additionally, protoplasmic astrocytes in humans propagate calcium waves at a faster rate. Fibrous astrocytes in the white matter of the human brain also have significantly larger sizes compared with other primates or rodents [83].

## 4. Oligodendrocytes

### 4.1. Morphological and Functional Heterogeneity

In the past, OLs were considered a homogeneous cell population. However, modern research indicates that they are highly heterogeneous not only in terms of their localization and morphology but also in terms of their genetic profile [85]. Regional and morphological differences in OLs were first identified in the 1990s. Based on their localization, three groups are distinguished: (i) interfascicular, (ii) perineuronal, and (iii) perivascular OLs. The best-studied and most typical representatives are the interfascicular OLs, which are found between axons in the white matter of the brain [85,86,87]. Experimental models utilizing genetic defects in laboratory mice are utilized to explore the morphology and functionality of oligodendrocytes across different brain regions. In mice where a key transmembrane protein characteristic of interfascicular OLs is genetically altered (knocked out), there’s a notable reduction in myelination within the spinal cord area. Interestingly, this effect is much more pronounced compared with the brain stem region. This observation suggests that distinct molecules and mechanisms are at play in different brain regions to execute the functions of OLs [85,88]. Perineuronal OLs, also known as “satellite” cells, surround the cell bodies of neurons in gray matter. They are considered “reserve” cells because, under certain pathological conditions (demyelination), they exhibit the ability to myelinate nerve fibers [89]. There is evidence that they influence neuronal excitability by participating in potassium metabolism [90]. Perivascular OLs come into contact with the blood vessels of the CNS, but their function is not fully understood. They have specific metabolic activity, suggesting a trophic role for nerve fibers [91].

Based on their morphology and the size/number of axons they myelinate, OLs are categorized into four subtypes (I–IV). Type I OLs have small, oval bodies and multiple cellular processes that myelinate a large number of axons with small diameters, distributed diversely throughout the brain tissue. They are found in both white and gray matter. Type II OLs are exclusively located in white matter of the brain, characterized by their cuboidal bodies and myelination of only parallel-oriented axons. Type III OLs are not as numerous as the other two types. They are found in both gray and white matter, similar to type I OLs. They have fewer cellular processes that myelinate axons with larger diameters. Type IV OLs, also known as “Schwann cell-like” OLs, myelinate only one axon with a large diameter, resembling Schwann cells in the peripheral nervous system. These cells have elongated bodies, parallel to the axons in white matter, where they are typically found [82,84]. These four types of OLs can be further subtyped based on their molecular profile and the difference in the length of the internodal segments along the axon they myelinate [92,93,94].

### 4.2. Identification Markers

The nuclei of OLs have specific morphological characteristics. Thanks to this, they can be easily distinguished from astrocytes and neurons in light microscopy preparations. Their nuclei are oval, relatively dark, and smaller compared with the nuclei of the mentioned cell types. The localization of OLs can be around three typical structures in the brain tissue, as already described.

### 4.3. Immunohistochemical Profile

A large portion of antibodies for detecting OLs are myelin-specific, such as myelin basic protein, galactocerebroside, and myelin-associated glycoprotein. However, they are not suitable for studying histological sections since the direct processes of oligodendrocyte bodies and processes cannot be traced in such sections. They are applicable in experiments with cell cultures [95].

### 4.4. Transcription Factors

Modern studies involving the analysis of transcription factors that regulate the processes of cell differentiation and proliferation in the CNS have proven to be of exceptional importance. The first identified transcription factors that modulate the development of OLs are Olig1 and Olig2 [96,97]. Another transcription factor that determines oligodendroglial identity is Sox10. There is a group of factors that are expressed transiently or during a specific stage of development and outline cell proliferation. These include chromatin-modifying enzymes and remodeling complexes such as microRNAs and long noncoding RNAs [98,99,100,101,102].

Olig2 and Sox10 are expressed at every stage of OL development, including mature forms [96,97,103,104,105,106]. Olig2 is expressed in neuroepithelial cells even before their differentiation into OL cell lineage [104,107]. Mature OLs express Olig2 with moderate intensity compared with OPCs or actively myelinating Ols [108]. Sox10 appears immediately after the differentiation of neuroepithelial cells and is present at a slightly later stage of OL development compared with Olig2. Therefore, the co-expression of Olig2 and Sox10 outlines the pre-OPC stage, which progresses to the OPC stage [109,110]. Once expressed, Sox10 exerts a significant regulatory effect during development and in the mature phase of OLs [111,112]. Olig2 and Sox10 mutually enhance their expression [113].

In most OLs, Olig1 is not essential for cell proliferation. In the absence of Olig1, the effects are partial or transient, depending on the affected region. Furthermore, the absence of Olig1 can be compensated by the presence of Olig2, while the absence of Olig2 is compensated minimally by Olig1 [104,105,108].

During OL development, Sox10 functions together with Sox8 and Sox9 proteins (gene paralogs). Co-expression of Sox10 and Sox8 is observed in OLs at every stage of development. Sox8 has minimal significance for cell progression. Sox9 additionally appears until the moment of the final differentiation of the cell lineage. Sox10 and Sox9 have equivalent functions, so both factors need to be deleted to affect OL development at a given moment [112,114,115,116,117,118]. Transcription factors expressed at various stages of OL development are presented in Figure 3. Considering the different characteristics of OPCs and mature OLs, it is assumed that these transcription factors perform different functions at different stages of cellular development. Their activity is regulated in a stage-specific manner, and this hypothesis is supported by experimental studies during specific periods of OL progression, where deletion or overexpression of Olig2 leads to different phenotypic manifestations [119,120]. OPCs are positive for NG2 and PDGFR, while the myelinating OLs are positive for markers such as O4 and proteolipid protein (PLP) [121,122].

### 4.5. Genetic Analysis

Genetic profiling analysis of Ols through RNA sequencing demonstrates that they represent a transcriptionally heterogeneous cell population. In 2015, Zeisel et al. conducted a study on the oligodendrocyte population in the somatosensory cortex and hippocampus of mice, revealing six OL types representing different stages of cellular maturation: postmitotic, immature, pre-myelinating, myelinating, intermediate, and terminally differentiated post-myelinating OLs [73]. A year later, Marques et al. performed additional analysis on over 5000 cells from the OL lineage in 10 regions of the CNS in young and adult mice, presenting a total of 12 OL lineage groups corresponding to different developmental periods from OPCs to mature OLs. While immature cell lineages were evenly distributed in the brain tissue, mature OLs showed specific regional distribution in the brain, with unique proportions in each area. Only young mouse brain tissue exhibited the entire spectrum of OL populations, while adult mice showed only OPCs and two types of mature Ols [125]. In 2018, Zeisel et al. conducted further investigations and identified the presence of 10 OL lineages but did not find different cell subtypes than those previously presented by Marques et al. in 2016 [126].

The first comprehensive characterization of OLs in humans was performed within a study investigating OL heterogeneity in white matter from five healthy individuals and four patients with multiple sclerosis. Clusters of OPCs and seven clusters of mature OLs, including a cell subtype expressing genes associated with immune reactivity, were identified [127]. Comparative analysis of the described cell subpopulations in experimental animals and humans revealed correspondences between the species [128].

To describe the complete spectrum of human OL subpopulations, Sadick et al. combined their experimental results with previously conducted studies and identified seven OL populations [129,130,131,132]. A similar approach was used to construct an atlas of cell types in the mouse spinal cord, but information regarding OLs was not sufficient [133].

The Brain Initiative Cell Census Consortium (BICCC) presented an atlas of the primary motor cortex in nonhuman primates, focusing primarily on the neuronal cell population. All other cell types (excluding neurons) were categorized into 26 clusters, of which 8 clusters were identified as OL populations [134].

A recent study from 2022 aimed to map regions of the human brain at different stages of development, including brain tissue from the postnatal period, adolescents, and young adults. The results indicate that the level of OL development in the spinal cord of adolescents is more advanced compared with the level of OL development in the cerebral cortex. Furthermore, mature OLs from the spinal cord, not microglia, express markers associated with immune reactivity. Superexpression of transcription factors related to development is observed in mature OLs from the subventricular zone (SVZ). Similar to results in experimental animals and humans, differences exist in the number of OLs and the extent of changes in the OL population during different developmental periods. These differences have a regional character. In brain tissue from young individuals, OL populations tend to selectively establish themselves in certain brain regions, and these preferences differ from the regions favored for proliferation in brain tissue from adults [125,135,136,137,138].

Based on gene expression, one could conclude that OPCs represent a homogeneous cell population [125], but as mentioned earlier, they populate different brain regions at different stages of development and can differentiate into diverse cell subtypes depending on the region they are located in. OPCs are also a heterogeneous cell population in terms of their proliferative potential. This characteristic is utilized in cell culture experiments [139,140,141,142]. There are hypotheses regarding the diversity of OPCs based on their functional activity in different areas of the CNS [142,143].

## 5. NG2-Glia

The glial cells expressing the chondroitin sulfate proteoglycan neuron-glial antigen 2 (NG2) are considered a distinct cell population in the CNS [144]. They should be distinguished from other non-glial cell types in the CNS that also express NG2, namely, pericytes around blood vessels [145] and macrophages, which temporarily become positive under certain pathological conditions [146]. Of interest is the fact that under physiological conditions and following brain injury, NG2-glia acquire the potential to differentiate into various cell types [21,29]. Other terms used to refer to these cells are “complex cells” [84], GluR cells [147], polydendrocytes, and synantocytes [148].

### 5.1. NG2-Glia: Homogeneous or Heterogeneous Cell Population?

The morphology of NG2+ cells has been well studied in the mouse hippocampus, where it has been found that these cells are positive for NG2, have clearly visible processes, and express enhanced green fluorescent protein (EGFP) with low intensity. They do not possess glutamate transport proteins and do not form gap junctions with each other.

On their surface, glutamatergic AMPA receptors (α-amino-3-hydroxy-5-methyl-4-isoxazolepropionic acid receptor), NMDA receptors, and GABA receptors can be found in different brain regions [30,149,150,151,152,153,154,155,156]. What distinguishes classical astrocytes is that they express EGFP with high intensity, lack AMPA receptors, transport glutamate through transporter proteins, and form gap junctions with other astrocytes [148,149,155,157]. Despite their morphological diversity, the electrophysiological properties of astrocytes in different brain regions are identical. However, NG2+ glial cells display heterogeneity in their electrophysiological characteristics [29,150,151]. They possess various ion channels on their surface, which enable fast cellular signaling and interaction with neurons. As mentioned earlier in Section 1, NG2-glia have different proliferative potentials in gray and white matter. Differences have also been observed in the duration of the cell cycle of NG2-glia, not only between gray and white matter but also among NG2+ cells in different white matter tracts within specific brain regions [156,158]. Heterogeneity can also be observed among NG2+ cells within the same brain region, where they express different transcription factors [159,160].

### 5.2. Interspecies Heterogeneity

Despite the difficulties in obtaining human brain tissue for experimental purposes, NG2-glia have been studied in the regions of the neocortex and hippocampus, and sufficient data have been accumulated to allow for a comparative analysis of NG2+ cells between humans and their rodent counterparts (Table 1) [29,30,149,150,151,161,162,163,164,165,166].

## 6. Microglia

Microglial cells are part of the immune-regulatory cells in the CNS. Although they are components of the glial populations in the brain tissue, their origin differs from that of macroglial cells, as described earlier. This cell population has the potential for self-renewal, so myeloid precursors do not need to pass through the BBB to maintain their cellular composition [167,168,169,170,171,172].

### 6.1. Morphology and Function

In human tissue from the frontal cortex and hippocampus, four morphological types of microglia have been described (Figure 4) [173]. In addition to their morphology, they also differ in their functions [174]. In their inactive state, microglia possess a specific phenotype: they have a small body and long ramifications. They are referred to as “ramified” microglia or microglia in a “resting state.” Their processes are dynamic and can retract and regenerate rapidly. Through these processes, microglial cells continuously “scan” the brain tissue to detect changes in homeostatic balance [175]. Microglia make contact with neurons, blood vessels, and other glial cells through their processes. They are responsible for maintaining neuronal function and synaptic plasticity in brain tissue through various biochemical interactions, including phagocytosis [175]. Microglia with phagocytic activity and the ability to present antigens are characterized by shortened ramification and a larger cell body. This is known as activated microglia, which perform their functions by releasing anti-inflammatory cytokines, growth factors, and neurotrophic factors, as well as by limiting the spread of certain pro-inflammatory cytokines [173,174]. However, under certain conditions, when the target structures are larger or unable to be engulfed, reactive oxygen species (ROS) are released from microglial phagosomes, which can be toxic [176,177]. Recent studies suggest that microglia phagocytose cellular debris is generated by apoptosis in regions with active neurogenesis in the adult brain. Additionally, microglia eliminate unnecessary synaptic elements through phagocytosis [173,175,178,179]. Characteristic biochemical interactions of microglial cells include the modulation of neurotransmission and the secretion of certain enzymes, such as matrix metalloproteinases (MMPs) and tissue plasminogen activator (tPA), which participate in the remodeling of the extracellular matrix in the synaptic area [175]. In pathological conditions such as intracranial bleeding and brain metastases, microglia transform into so-called amoeboid microglia with rounded cell somata, no ramification, and phagocytic activity [174,180]. In the aging human brain, a fourth category of microglial cells has been identified. These cells have lost their capacity for phagocytosis and neuronal support. Morphologically, they are marked by swelling and thinning of their processes (appearing fragmented) and a loss of ramification. Due to this reason, they are designated as dystrophic microglia [174,180].

### 6.2. Molecular Heterogeneity

Subtyping microglia is a challenging task due to overlapping gene expression patterns with many cell types. Furthermore, some characteristics are identical to those of macrophages that infiltrate the CNS from the peripheral circulation. Nevertheless, several specific proteins and genes characteristic of the microglial cell population have been discovered. Immunofluorescence studies have demonstrated the expression of ionized calcium-binding adapter molecule 1 (IBA1), transmembrane protein 119 (TMEM119), and the purinergic receptor P2Y12 (P2RY12), as well as a range of cell differentiation markers from the CD family, such as CD16, CD68, and CD11b [181,182,183,184,185]. ScRNA seq has also been used to identify microglia using differential expression of specific genes in a similar manner [183,184].

The results of a study by Zheng et al. indicate the existence of different microglial subtypes in the wt mouse spinal cord. However, there is only partial overlap in the genetic profiles of cells in the spinal cord compared with the cerebral cortex [186]. Another experiment, which analyzed microregions in the cerebellum, cerebral cortex, hippocampus, and striatum of mice, also provides information on the presence of different subtypes of microglial cells based on their gene expression [187]. However, a third study aiming to investigate microglial subpopulations in the same regions of mice through single-cell deep sequencing did not confirm the previously known results [188], indicating the need for additional experimental data to accurately describe microglial heterogeneity.

## 7. Ischemic Brain Injury

### 7.1. Clinical Significance

The term “stroke” refers to a sudden loss of neurological function resulting from acute vascular disruption of cerebral blood flow. Approximately 80% of strokes are clinically manifested as cerebral infarction. Stroke is the second leading cause of death and the primary cause of physical disabilities in adults worldwide. In Bulgaria, there are 35,311 registered cases annually, of which 7175 result in fatalities. The number of survivors with varying degrees of disability is 28,136, with 25–50% experiencing speech and language disorders [189].

### 7.2. Forms of Brain Ischemia

There are two main pathomorphological forms of cerebral infarction. The so-called anemic or pale infarction is characterized by a disruption of arterial blood flow in the corresponding area. The brain tissue in this region remains poorly perfused and becomes pale. After hemolysis of erythrocytes in the affected area (after 48 h), the infarct becomes yellowish-white. In the case of hemorrhagic infarction or red brain softening, there is initially an obstruction of an arterial vessel by a thrombus of sufficient duration, leading to the development of anemic infarction. Subsequently, the thrombus lyses or partially fragments, blocking distal, smaller branches of the respective artery. Blood flow is restored in the ischemic area, resulting in the seepage of blood into the zone. As a result, the necrotic area takes on a dark red-to-black color. The middle cerebral artery or its branch is most commonly affected. Hemorrhagic infarction of brain tissue can also occur with extensive thrombosis of the sagittal venous sinus [190]. Thrombosis of small arteries and arterioles leads to so-called lacunar infarcts, which are multiple and involve the simultaneous blood supply of several brain territories. Multifocal infarction is observed, often with hemorrhage [191].

Global cerebral ischemia develops in the context of severe hypotension or increased intracranial pressure. Although cerebral infarction is most commonly caused by ischemia (ischemic hypoxia or stagnant hypoxia), it can also result from a reduced amount of oxygen in the blood without a reduction in cerebral blood flow (hypoxia or anemic hypoxia), or it can be the result of toxins that prevent cells from using oxygen for oxidative processes (histotoxic hypoxia). In addition to oxygen deficiency, the brain tissue is also highly sensitive to glucose deficiency (hypoglycemia) [191].

### 7.3. Etiology, Pathogenesis, and Morphology

The causes of cerebral infarction are numerous and can generally be represented as vasculopathies and vasculitis of large and small blood vessels. These conditions can manifest clinically in several ways, but the most common manifestation is infarction of the brain tissue.

Atherosclerosis of large blood vessels most commonly occurs extracranially and can be a source of emboli in small cerebral arteries, such as thrombus or atheromatous material. Less frequently, thromboembolism is a complication of arteritis or vascular aneurysms. Infectious vasculitis in the context of septicemia or in patients with immunodeficiency most commonly affects small arteries, arterioles, and venules, leading to thrombosis with infarction. Other rarer causes of embolism include fat embolism, air embolism, tumor embolism in malignant diseases, or embolism from an intervertebral disc following trauma. Disorders in the coagulant-anticoagulant system, various etiologies of vascular malformations, and vascular tumors are also among the causes of cerebral tissue infarction.

Saccular berry aneurysms, which affect smaller branches of the major cerebral arteries, can cause cerebral hemorrhage. Very rarely, small intracranial arteries can be affected following the dissection of a large blood vessel, such as the carotid artery. Depending on the affected blood vessel and the presence or absence of collateral vessels, the infarction can be limited beneath the pia mater, be deeper, or involve the entire zone supplied by the affected vessel [191,192,193]. Macroscopically, within the first 24 h, the infarcted gray brain matter becomes reddish-brown due to pronounced vascular congestion. Focal petechial hemorrhages may be present. The white brain matter also darkens in color and is congested, but the changes are more discrete. The boundaries of the infarcted area are difficult to determine earlier than 2 days after the incident. After this period, the affected tissue is significantly softer compared with the surrounding healthy parenchyma. After several days and weeks, the affected tissue undergoes liquefactive necrosis, and a cyst forms.

Histologically, at 4–6 h after the incident, neurons in the affected area become distorted and have pyknotic nuclei, perineuronal glia appears swollen with pale cytoplasm, and microvacuolization is observed among the neuropil. Between 6–48 h, the cytoplasm of neurons becomes increasingly eosinophilic, and the nuclei become amphophilic or also eosinophilic. After 2–3 days, it is difficult to identify neuronal nuclei, although their cytoplasmic boundaries are clearly visible. Glial cells, especially perifascicular and satellite OLs, undergo apoptosis and can be identified as apoptotic bodies within the infarcted tissue. Neutrophilic leukocyte infiltration is observed at the periphery of the infarct at 6–12 h. Between 1 and 2 days, the inflammatory reaction is abundant, but mononuclear cells predominate. Foamy macrophages accumulate in the infarcted area and around blood vessels. Neovascularization of the necrotic tissue occurs as a result of endothelial proliferation. After several months, the necrotic tissue is resorbed, and a pseudocyst lined with glial cells remains at the site of the incident, with blood vessels in the wall and residual foamy macrophages. Some of the foamy macrophages may contain hemosiderin. Numerous reactive astrocytes are observed in the surrounding parenchyma [191,192,193].

### 7.4. The Role of Macroglia in Ischemic Brain Injury

#### 7.4.1. Reactive Astrogliosis and Its Role in Ischemic Brain Injury

“Reactive astrogliosis” is a term used to describe cellular, functional, and molecular changes that astrocytes undergo following injury. When morphological changes have occurred, they can vary from hypertrophy of cell bodies and processes to alterations in protein profiles and/or proliferative activity [194,195]. The extent of observed changes depends on the severity of the injury. Therefore, astrocytic reactivity can be characterized as mild, moderate, diffuse, or severe. This quantitative analysis is based on the degree of GFAP expression. The more pronounced the astrocytic transformation and consequently the severity of the injury, the stronger the intensity of GFAP expression [194,196]. Several intercellular and intracellular signaling molecules regulate this process. The effect can be both protective and exacerbating the damage. Reactive astrocytes provide neuroprotection in the acute stage of ischemic brain injury while simultaneously interacting with immune cells from the blood, endothelial cells, and microglia, leading to the development of brain edema and potentiation of the neuroinflammatory response. In the chronic stage, reactive astrocytes are responsible for the formation of a glial scar, which aims to limit the damage zone, tissue remodeling, and restoration of neuronal functions [192,193,197].

Two subtypes of reactive astrocytes have been described: A1 and A2 reactive astrocytes. The proliferation of the A1 subtype is stimulated by secreted IL-1α, TNFα, and C1q from activated microglia, leading to neuronal and OL death. A2 reactive astrocytes have a neuroprotective effect and secrete trophic factors for neurons [198]. Among the markers expressed in reactive astrocytes are Lcn2, GFAP, Vimentin, and Timp1. Transcriptional analysis of reactive astrocytes in experimental animals following ischemia confirms the claim that this cell type, similar to microglia, has both pro-inflammatory and neuroprotective functions [199,200].

High-tech imaging methods have been used to study the ischemic penumbra in the human cerebral cortex [201]. Changes in astrocyte morphology can be described according to the stage of cerebral infarction. In the acute phase (1st to 4th day after ischemia), astrocytes exhibit increased proliferative activity and increased GFAP expression. In the subacute phase (4th to 8th day after ischemia), astrocytes with elongated processes and depolarized cell membranes are described, gradually forming a glial scar until the onset of the chronic stage (8th to 14th day after ischemia). Differences in astrocyte reactivity are associated with their sensitivity to ischemia, their location relative to the lesion core, and their subtypes [202]. Some authors believe that the different reactivity is also due to differences in the protein profile [203].

In ischemic brain injury, the lack of glucose is compensated by reactive astrocytes, which can initiate the process of glycolysis to produce lactate, serving as an energy source for neurons and transporting it to them through specialized transporters. Data regarding this function of reactive astrocytes and their role in the survival or degeneration of neurons are contradictory [204].

By establishing the connection between neurons and blood vessels within the neurovascular units in the CNS, astrocytes play a key role along with individual cells and pericytes in the aftermath of cerebral tissue infarction. Reactive astrocytes secrete VEGF (vascular endothelial growth factor) and MMPs, which increase vascular permeability and exacerbate ischemic damage in the acute phase of the incident. In the chronic phase, when the process of neuroregeneration prevails, the same bioactive molecules stimulate angiogenesis and the restoration of the BBB [205,206].

Excessive glutamate in the extracellular space of brain tissue has neurotoxic effects. This may be due to the inability of astrocytes to eliminate it through reuptake and convert it into glutamine. Therefore, by participating in the glutamate–glutamine cycle, reactive astrocytes have a protective effect on neurons [67,207]. Reactive astrocytes also secrete synaptic molecules, such as cholesterol-associated apolipoprotein E (APOE) and thrombospondin [208,209].

Astrocytes exert their antioxidant action through the production of glutathione, which is important for limiting the damage during ischemia in the cerebral cortex [210]. However, reactive astrocytes can also release ROS and nitric oxide, leading to oxidative stress [176]. Recent studies have shown that astrocytes can produce extracellular vesicles containing proteins, lipids, and nucleic acids and thus interact with other cell types. There is evidence that under ischemic conditions, reactive astrocytes can increase the survival of neurons through such vesicular activity [211,212,213]. However, studies on human astrocytes have demonstrated that these types of vesicles can be perceived by neurons and have adverse effects on their functioning and differentiation [214].

A significant portion of the mitochondria in the axons of retinal ganglion cells is normally degraded by astrocytes in the optic nerve head (ONH). This transcellular process of mitochondrial degradation, known as transmitophagy, likely occurs in other regions of the CNS as well, as structurally similar accumulations of degrading mitochondria are observed along neurites in the superficial layers of the cerebral cortex [215]. The introduction of astrocyte mitochondria into neighboring neurons has a protective effect during temporary focal cerebral ischemia in mice [216]. This raises the question of whether astrocytes in adult brain tissue can transfer mitochondria into affected neurons after ischemic injury [217]. During ischemic brain injury, astrocytes exert neuroprotective and anti-inflammatory effects, attributed to the secretion of erythropoietin, VEGF, GDNF (glial cell line-derived neurotrophic factor), and estrogen (17β-estradiol), which limit neuronal damage [218,219,220].

Gap junctions between astrocytes remain open during in vivo ischemia and in vitro hypoxia. This allows for the passage and rapid spread of pro-apoptotic factors, contributing to an increase in the size of the necrotic area. However, there is evidence from experiments on animals suggesting that astrocytic gap junctions can limit the zone of necrosis. The exact role of these contacts is still contradictory, and further research is needed to clarify their significance [206,221].

There is a hypothesis that reactive astrocytes forming a scar are actually astrocyte-like neural stem cells that differentiate into astrocytes. This transformation is thought to be modulated by specific genes activated after ischemia. Reactive astrocytes isolated from the peri-infarct cortex following ischemia can de-differentiate into neural-sphere-producing cells (NSPCs), which are multipotent cells capable of self-renewal. However, when transplanted, these cells have been shown to differentiate into astrocytes and OLs, but not neurons. Nevertheless, this demonstrates the high plasticity of reactive astrocytes. Recent studies indicate that reactive astrocytic glial cells after ischemia can be reprogrammed into functioning neurons, leading to a reduction in gliosis and restoration of synaptic contacts. There is also evidence that a combination of transcription factors can transform reactive astrocytes not only into neurons but also into neuroblasts. This highlights once again the plasticity of reactive glia and the potential for this property to find applications in targeted therapies following ischemic brain injury [206,222].

#### 7.4.2. Role of Oligodendrocytes in Ischemic Brain Injury

OLs are highly susceptible to ischemia, and a significant number of them die within three hours of an acute incident [223]. However, they play a crucial role in the chronic stage of ischemia as the main cellular population responsible for remyelinating affected axons [224]. After ischemia, mature OLs accumulate along the border of the infarct zone to participate in tissue recovery [225].

It should be noted that OLs do not have the capacity for self-renewal. In fact, ischemia stimulates the proliferation and differentiation of OPCs into myelinating Ols [226]. The number of OPCs increases in the penumbra (the surrounding region of the infarct) following ischemic brain injury but decreases in the center of the lesion. They undergo morphological transformation characterized by hypertrophy of cell bodies as well as molecular and genetic changes that stimulate their migration, proliferation, and differentiation [227,228,229,230,231].

During ischemic brain injury, OLs undergo apoptosis induced by the complement system and the toxic effects of released glutamate and ATP. Additionally, OLs are influenced by inflammatory cytokines primarily released by microglia in the infarct area. For example, IFN-γ induces apoptosis, delays remyelination, and inhibits the proliferation and differentiation of OPCs. TNF-α also induces apoptosis and delays remyelination. IL-6, IL-11, and IL-17 have a beneficial effect by promoting the survival of OLs. IL-1β has contradictory effects: it promotes the survival of OLs on the one hand and contributes to their necrotization on the other hand [232]. Numerous studies indicate that interactions between microglia and OLs can have both favorable and unfavorable effects, depending on the stage of OL development [233].

#### 7.4.3. Role of NG2-Glia in Ischemic Brain Injury

NG2-glia rapidly respond to ischemic damage in brain tissue. Together with macrophages, these glial cells are identified within the first day after the onset of injury [195]. The observed changes in NG2-glia are morphological, including hypertrophy of the cell body, shortening and thickening of cellular processes, and increased intensity of NG2 expression [234]. Their proliferative activity is also enhanced, and NG2+ cells migrate to the periphery of the ischemic lesion [195]. After this stage, the number of NG2-glia gradually decreases, reaching optimal levels within 28 days after the injury. This suggests that these glial cells perform a regulatory function and maintain homeostatic balance following the injury [158].

Experimental studies conducted on mouse brains after focal cerebral ischemia reveal the presence of cells containing genes characteristic of both NG2-glia and reactive astrocytes. Immunohistochemical analysis of their protein profile confirms these characteristics. Consequently, they are referred to as astrocyte-like NG2-glia cells. Their profile resembles that of astrocytes in the cortical gray matter. They are localized in the postischemic glial scar and are likely related to its formation following ischemia [158,235,236]. Additionally, another study demonstrates the involvement of NG2-glia in the early stages of tissue recovery after brain injury [237].

### 7.5. Influence of Other Factors in Ischemic Brain Injury

#### 7.5.1. Role of Microglia

Upon the appearance of a damaging factor, microglia undergo a transformation from a resting state to what is known as activated or amoeboid microglia [180]. These cells have undergone the following morphological changes: hypertrophy of the cell body and retraction and thickening of cellular processes [238]. Several markers are predominantly expressed in activated microglia, including CD45, MHCII, and CD68 [239]. Iba1, IB4, F4/80, and CD68 markers are also used for their study [181,240].

Following focal cerebral ischemia, different microglial phenotypes are observed depending on the location of microglia relative to the site of injury and the specific expression of surface cell proteins. In the periphery of the infarct, microglial cells are positive for Iba1 and negative for CD68. In the center of the infarct zone, cells are positive for both Iba1 and CD68, but they also show increased expression of CD11b [181,240]. While the activated microglia in the infarct zone exhibit phagocytic activity and primarily express MHCI [241], microglia in regions remote from the lesion express MHCII and are associated with neuronal degeneration [242,243]. These diverse morphological subtypes of microglial cells are presumed to perform different functions depending on their distance from the ischemia and the time elapsed since the incident [240,244,245,246,247].

Microglia may participate in the reconstruction of blood vessels following ischemia. They carry out this function through the phagocytosis of endothelial cells and the release of the pro-angiogenic VEGF [248]. Endothelial proliferation, which represents the initial step of angiogenesis, is influenced by various pro- and anti-inflammatory cytokines, with TGF-α stimulating it and TGF-β inhibiting it [249]. Microglia with an anti-inflammatory effect, referred to as the M2 subtype, secrete TGF-β and are predominantly located in the ischemic area during the acute and subacute phases of cerebral infarction [250]. M2 glia also secrete pro-angiogenic factors such as VEGF and matrix metalloproteinase-9 (MMP-9) [251]. It has been demonstrated that following a stroke, the integrity of the BBB is maintained by pro-inflammatory factors such as TNFα, IL1-β, and IL-6, released by the so-called M1 microglia [252]. While resting microglia create an environment that inhibits endothelial proliferation, activated microglia have the opposite effect [249].

According to several studies, ischemia stimulates neurogenesis in both rodents [253] and primates [254]. In the ipsilateral SVZ, activated microglia with cellular processes facilitate the migration of neuronal precursor cells, while amoeboid microglia in the peri-infarct zone may have an adverse effect on neurogenesis [255].

Neuronal damage in ischemic brain injury enhances the phagocytic activity of microglia. Activated microglia have a dual role: on the one hand, they support neuronal survival [255], but on the other hand, uncontrolled microglial activation can lead to the release of cytotoxic factors such as superoxides [256,257,258], nitrogen oxide [259] and TNF-α [260]. NADPH oxidase is a mediator with neurotoxic effects and is released during ischemic brain injury. It stimulates microglia, which in turn causes long-term inhibition of synapses [261]. This means that the function of neuronal circuits after ischemia can be influenced by microglia.

Advances in vivo imaging techniques make the study of microglia in humans accessible. Activated microglia can be observed as early as 24–48 h after ischemic stroke, mainly localized in the center of the injury and gradually spreading to the peripheral [262,263]. Therefore, this cell type performs its function in both the acute [262,263] and subacute phases of brain infarction [264,265]. Clinical studies involving six patients within a period of 3 to 150 days after diagnosis of ischemic stroke confirm these findings: activated microglia can be observed in the early days of ischemia, distributed in the infarct area and gradually spreading toward the periphery. Reactive microglial cells are also present in the contralateral hemisphere at a later stage [266].

According to another study involving four patients, activated microglia are present in both the infarct zone and its surroundings during the first week after the incident, but the number of these cells gradually decreases over time. After 14 weeks, it has been observed that the increased amount of reactive microglia persists in the peri-infarct region, while it is lower in the center of the lesion [265]. In 18 patients with a first ischemic stroke in the subcortical region, different clinical outcomes were observed depending on whether activated microglia were mainly localized in the infarct zone or its surroundings. The effect of activated microglia in the core of the lesion is negative, while the activation of microglia in the peri-infarct region correlates with a positive clinical outcome [267]. The available contradictory data from human studies show that the question of whether reactive microglia have a positive or negative effect on recovery after ischemic stroke is still debatable.

#### 7.5.2. Interactions between Glial Cells

During cerebral infarction, the integrity of the BBB is disrupted, which critically affects the extent of brain damage. Bioactive substances released by endothelial cells, brain glial cells, and immune cells provoke and sustain the inflammatory reaction in this area. Among the brain cell populations, microglial cells are the first to respond to the injury, activating through a series of molecular mechanisms and transforming into different functional subtypes. This is followed by infiltration of immune cells and a similar reaction by astrocytes. Microglia have the ability to modulate astrogliosis following brain ischemia, as reactive astrocytes possess a range of receptors for signaling molecules secreted by microglia. It can both stimulate and limit astrogliosis depending on specific conditions. On the other hand, microglia influence neurotransmission controlled by astrocytes by stimulating the release of glutamate into the extracellular space [206,268,269].

Astrocytes, in turn, can actively modulate microglial activity, both locally through paracrine mechanisms and at distant sites through mediators such as IL1β, the calcium signaling pathway, ATP, and the cytoplasmic calcium-binding S100β protein, and through the production of gliotransmitters, inflammatory cytokines, and RNA molecules. There is evidence that astrocytes can also influence gene expression in microglial cells [206,269].

The elimination of degenerated neurons resulting from ischemia is a process in which microglia and astrocytes interact once again. Experiments with mice modeling chronic cerebral hypoperfusion demonstrate characteristic cell interactions. The bodies of damaged neurons are infiltrated by processes of both astrocytes and microglia, forming a complex structure known as a triad, which exacerbates ischemic damage and contributes to an increased degree of neurodegeneration [270].

Reactive astrocytes secrete a range of bioactive molecules that are relevant to the functioning of OL cell lines [206]. Additional information is presented on Table 2.

The integrity of astrocyte-oligodendrocyte contacts is relevant to the proper functioning of OLs. Defects in gap junction contacts and the proteins that mediate them, both in astrocytes and OLs, primarily have a negative effect and lead to disruptions in axonal myelination [206]. CX43 is an astrocytic protein from the connexin family that participates in the formation of gap junction contacts. An interesting discovery is that in cell cultures of astrocytes and OLs under hypoxic conditions, where CX43 is blocked, OPC differentiation is facilitated [271].

Activated microglia are harmful to OL progenitor cells but improve the survival of mature Ols [233]. A recent study demonstrates that Iba1+ microglia have a beneficial effect on the OPC subpopulation in the early stages following an acute ischemic incident but lose this pro-regenerative potential in later stages. Immediately after ischemia, they contribute to increased OPC density and limit the demyelination process. In vitro, experiments have demonstrated the direct beneficial effect of microglial vesicles on the same OPC subpopulation [247].

## 8. Role of Transcription Factors in Ischemic Brain Injury

The precise regulation of neural progenitor cell (NPC) proliferation and their differentiation into neurons, and later into macroglial cells (astrocytes and OLs), during embryonic development is crucial for the fate of the CNS. Several transcription factors serve as regulators that directly induce the differentiation of RGSCs into glial cells. At the same time, accumulating evidence suggests that the known signaling molecules and regulatory pathways that modulate glial differentiation are not key determinants [117].

### 8.1. Transcription Factor SOX10

SOX10 is a transcription factor belonging to the SOX family, which consists of 20 transcription factors with a similar DNA-binding domain called the “high mobility group (HMG) box.” Each of them is associated with the sex-determining region Y (SRY), a transcription factor found on the Y chromosome in males that plays a key role in male sex determination [123]. The SOX family contributes to cellular differentiation in various processes during embryonic development, such as sexual differentiation, skeletogenesis, neurogenesis, and neural crest development [272]. SOX10 regulates the differentiation of OLs and is expressed throughout the OL cell lineage, from OPCs to mature OLs, both during embryonic development and in the adult brain tissue. As the cell matures, Sox10 is expressed more intensely [111,112]. Additionally, by interacting with factors responsible for astrocyte differentiation, SOX10 can inhibit astrocyte differentiation. Therefore, it is a factor that determines the fate not only of OPCs and their progeny but also of astrocytes [273]. Furthermore, there is evidence of existing SOX10+ astrocytes in the hypothalamus and cerebral cortex [274]. A study demonstrates how SOX10 can transform satellite glial cells into OL-like cells in the peripheral nervous system [275]. It is known that in vivo, SOX10 can transform mature astrocytes into OPC-like cells that exhibit an oligodendroglial phenotype [276]. Additionally, under pathological conditions, differentiated OLs can transdifferentiate into astrocytes [277] and vice versa [276].

### 8.2. Transcription Factor SOX9

SOX9 plays a role in CNS development. It is expressed in NSCs during embryonic development and regulates the processes of differentiation in both astrocytes and OLs. According to several studies, SOX9 interacts with other genes and regulates them. It acts as a stimulus for transcriptional cascades that coordinate the development of glial cells [117]. Sun et al. conducted a study showing that SOX9+ cells are present in the neurogenic regions of the adult brain. They demonstrate the lack of expression of this transcription factor in OPCs, suggesting that these positive cells are most likely neural stem cells and/or progenitor cells at different stages of development, indicating the involvement of SOX9 in neurogenesis [12]. The same research group found that SOX9 is not expressed in mature neurons but serves as a specific marker for mature astrocytes outside the neurogenic niches of the CNS in mice. The new conclusion is that even if SOX9 regulates the processes of differentiation in both astrocytes and oligodendrocytes, its expression is astrocyte-specific. The quantity of SOX9+ glial cells does not change with age, but the number of SOX9+ reactive astrocytes increases in the symptomatic stage following spinal cord injury. Moreover, overexpression of SOX9 is observed in the penumbra following middle cerebral artery occlusion in mice, as well as in lacunar infarctions of brain tissue [12,112]. Furthermore, Sun et al. confirm that SOX9 is expressed in mature astrocytes in the cerebral cortex of humans and the transcription factor as an astrocytic-specific marker [12].

### 8.3. Transcription Factor ZBTB20

ZBTB20 is a transcription factor with specific significance in the CNS. It belongs to a group of transcription factors that contain the BTB/POZ (broad complex tramtrack bric-a-brac/poxvirus and zinc finger) domain. It is a regulatory DNA-binding protein that functions as a transcriptional repressor. Its action leads to epigenetic changes in cells during embryonic development and various pathological conditions, including neoplastic processes [11,278,279]. Other names used for ZBTB20 include HOF, Zfp288, ODA-8S, and PRIMS [280,281]. In humans, the ZBTB20 gene is located on the long arm of chromosome 3 (3q21) while in mice, it is on chromosome 16. It possesses an N-terminal domain responsible for protein–protein interactions and homodimerization, as well as a C-terminal domain consisting of five C2H2 zinc fingers organized into three clusters, which participate in binding to specific DNA sequences in the promoter regions of target genes. Due to the presence of a specific domain in the C-terminal region, ZBTB20 has an affinity for binding to DNA regions rich in adenine-thymine (AT) nucleotides. It is within this C-terminal region of the gene that the transcriptional repression domain is located, which is crucial for the role of ZBTB20 in regulating gene expression during neuronal development and functioning [278,281,282].

#### 8.3.1. Expression of ZBTB20 in the CNS in Rodents and Humans

During the embryonic period in mice, Zbtb20-positive cells include progenitor cells involved in astrocytogenesis and oligodendrocytogenesis, and it has been demonstrated that Zbtb20 is a key regulator of astrocyte development [11,283]. Zbtb20 is also expressed in developing hippocampal neurons and plays a role in hippocampal development and function [284]. In vitro and in vivo experiments with genetically modified mice (with suppressed expression or overexpression of Zbtb20) indicate that Zbtb20 stimulates astrocytic proliferation and inhibits neuronal proliferation in the neocortex [11].

In adult rodents, temporary expression of Zbtb20 is observed in pyramidal neurons of the cerebral cortex in Layers II–III, which later disappears. From a functional perspective, it has been demonstrated that Zbtb20 modulates the laminar generation of neurons in the cerebral cortex [284]. Similar to mice, Zbtb20 is also expressed in hippocampal pyramidal neurons during the embryonic period and in adults in both mice and humans. Its expression in macroglial cells persists throughout life, but both its expression and the role of this transcription factor as a regulator of macrogliogenesis in the cerebral cortex in humans are not well studied and require additional research [281]. Zbtb20 has been shown to be present in GFAP+, S100β+, or Aldh1L1-GFP+ astrocytes in the forebrain and hippocampus, as well as in the gray and white matter of the spinal cord in rodents [11,285]. Other areas of the CNS containing Zbtb20+ cells include the postnatal olfactory bulb (in astrocytes and glomerular interneurons) [283], the hypothalamus (in neurons of the suprachiasmatic nucleus) [286], the cerebellum (in immature and migrating granule neurons) [281,287], and in the peripheral nervous system; ZBTB20 is found in developing dorsal root ganglia and mature sensory bipolar neurons [280,288,289], as well as in the subventricular zone of the telencephalon [283].

#### 8.3.2. The Role of ZBTB20 in Ischemia

The influence of ZBTB20 on neurogenesis and gliogenesis processes has been studied in the mouse brain. In the intact brain of adult animals, the number of ZBTB20-expressing cells is increased in the SVZ region. Some of these cells are actually GFAP-positive radial glial cells, including NSCs. The latter are bipotent and can differentiate into both astrocytes and OLs. In animals with a completely suppressed ZBTB20 gene (homozygous loss of function), the number of GFAP+ cells is reduced, with the remaining ones mainly found in the SVZ. The number of S100β+ cells in the gray matter of the neocortex is also decreased. The expression of NG2 in glial cells in white matter regions (near the rostral migratory stream) and Olig2+ cells in the olfactory bulb is also reduced. This indicates that gliogenesis is generally suppressed in ZBTB20 mutant mice shortly after the transition from neurogenesis to gliogenesis. ZBTB20 likely regulates this process. Following induced ischemic brain injury, the number of ZBTB20-positive cells increases. In the ipsilateral SVZ (the side affected by ischemia), this increase is more significant compared with the contralateral side. The number of ZBTB20+ cells is also increased in the ischemic zone itself, and almost all of them co-express ZBTB20 and GFAP. However, some individual GFAP+ cells remain negative for ZBTB20. Meanwhile, in transgenic mice with reduced levels of ZBTB20 (heterozygous loss of function, where only one allele of the ZBTB20 gene is suppressed), the size of the glial scar following ischemic brain injury is decreased. This supports the hypothesis that ZBTB20 is a regulator of reactive astrogliosis in pathological conditions, including after ischemic brain injury [11,283]. Additionally, a study from 2016 identified ZBTB20 as a risk gene contributing to the development of ischemic stroke [290]. Similarly, a study conducted on intact brains of adult monkeys also found co-expression of ZBTB20 and GFAP in the SVZ and rostral migratory stream (RMS). In the brain of monkeys after ischemic injury, increased expression of ZBTB20 mRNA signal in the SVZ was observed. These results correspond to the findings obtained from rodent experiments. Therefore, ZBTB20 can be considered a regulator of precursor cell proliferation in the SVZ of primates. Whether it is involved in modulating gliogenesis processes and plays a role in the transition from neurogenesis to gliogenesis in primate brains remains to be further investigated [291].

## 9. Conclusions and Future Perspectives

Based on the information presented thus far, it follows that the adult primate brain possesses a diverse array of mechanisms for regeneration and tissue restoration. Throughout one’s lifespan, the brain undergoes dynamic changes in its cellular populations. Moreover, pathological conditions, including cerebral damage resulting from ischemia, impact the processes involved in the regeneration and recovery of nervous tissue, even stimulating these processes to some extent. Consequently, it becomes imperative to accumulate further data regarding the postnatal development of the human brain at the molecular level. A detailed comprehension of the structure of the adult mammalian brain, coupled with an intricate mapping of its cellular composition, would provide clarity concerning the subsequent alterations that occur following brain damage. Regrettably, the recovery of both neurons and glial cells remains an area of active investigation. We are now presented with the opportunity to modulate the plasticity of nervous tissue, with the aim of treating various pathological conditions of the CNS. In addition to addressing the aforementioned questions, this review also emphasizes the detailed study of the pathological processes associated with cerebral ischemia, as well as the role played by certain genetic factors in determining the fate of glial brain cells.

## Figures and Tables

**Figure 1 biomedicines-11-02332-f001:**
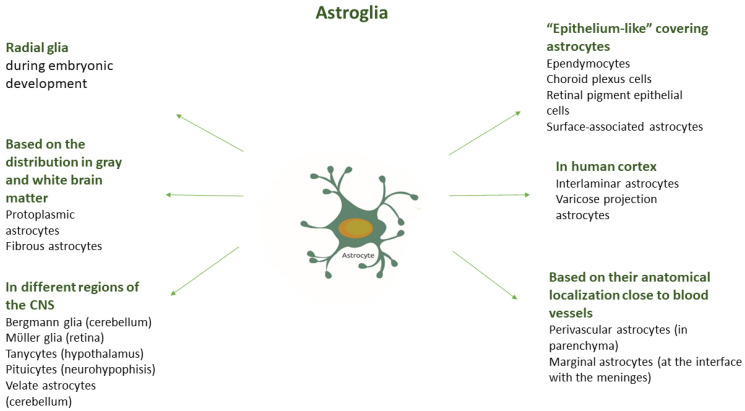
The scheme illustrating different subtypes of astrocytes [32,54].

**Figure 2 biomedicines-11-02332-f002:**
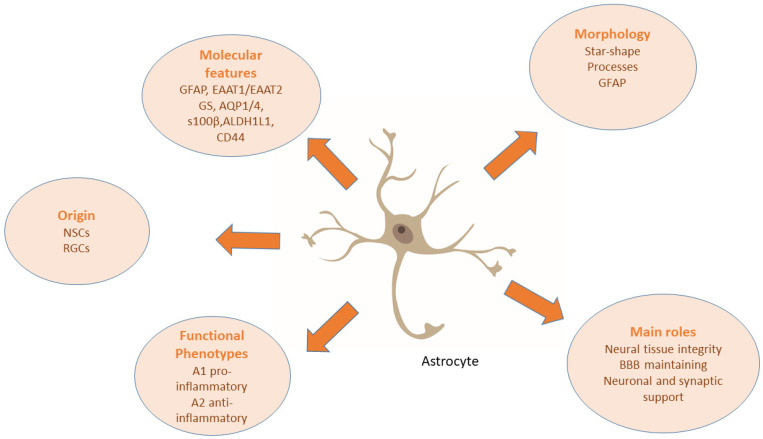
Astrocytic heterogeneity and the molecular profile of astrocytes. Illustrated molecular features include those that are discussed in the current review.

**Figure 3 biomedicines-11-02332-f003:**
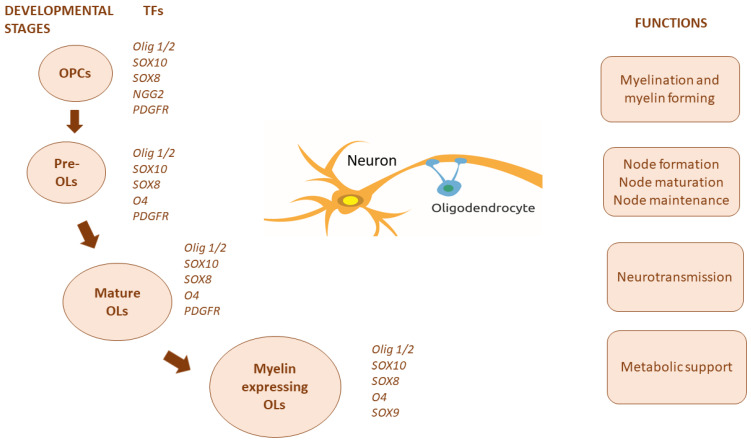
Аntigenic properties of OPCs, pre-OLs, mature OLs, and myelinating OLs as well as functions of OL cell lineage. The list of markers specific to each stage is limited to those referenced in the current review [96,108,112,114,118,123,124].

**Figure 4 biomedicines-11-02332-f004:**
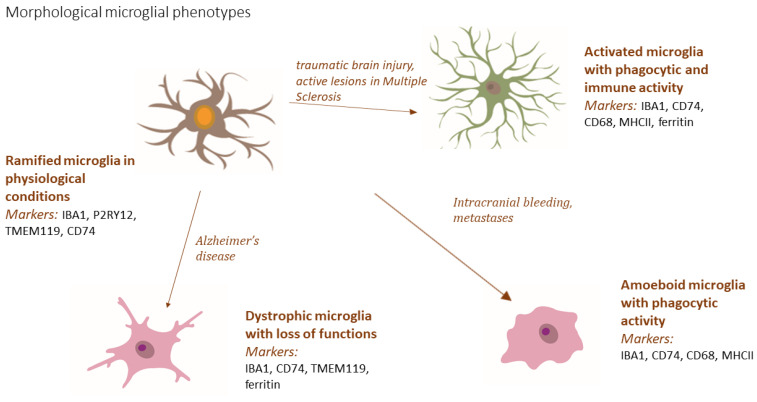
Alterations in microglial morphology [173].

**Table 1 biomedicines-11-02332-t001:** Comparative qualitative and quantitative analysis of NG2+ macroglial cells in the mouse and human hippocampus [29,30,149,150,151,161,162,163,164,165,166].

Parameters	NG2-Glia in Mouse	NG2-Glia in Human
Gray matter localization	Stellate shape, centrally round body, and several peripheral processes	Fewer branches compared with astrocytes’ larger bodies, fewer processes
White matter localization	Elongated body and processes oriented parallel to axons	Larger bodies, fewer processes
Gap junctions	No	No
Co-expression of NG2 and platelet-derived growth factor-alpha (PDGFRα)	Yes	Yes
Expression of GFAP	No	No
Expression of S100β	In a portion of the cell population	In almost the entire cell population
Electrophysiological properties: membrane characteristics, expression of ion channels, segregated expression of glutamate receptors and transporters	Similar	Similar

**Table 2 biomedicines-11-02332-t002:** Secretory activity of reactive astroglial cells following ischemic brain injury and its influence on oligodendrocyte cell line functions [206].

Reactive Astrocytes—Secreted Biomolecules	Effect on Olgodendrocyte Population
TNF-α, IFN-γ, IL-1β	Induce apoptosis and hypomyelination
IGF-1, EPO	Stimulate oligodendrogenesis
FGF2, PDGF	Stimulate oligodendrogenesis
FGF-2	Stimulate OPCs proliferation while inhibiting differentiation into OLs
CNTF in fibrous astrocytes	Stimulate migration of OPCs from SVZ to hypomyelinated regions
Bone morphogenic proteins (BMPs)	Block OPCs proliferation; stimulate differentiation of OPCs into astrocytes

## Data Availability

Data sharing not applicable. No new data were created or analyzed in this study. Data sharing is not applicable to this article.

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
