# Peer review of "Glial Populations in the Human Brain Following Ischemic Injury"

_biomedicines, 2023, doi:10.3390/biomedicines11092332_

Round 1

Reviewer 1 Report

First of all, I want to congratulate the authors who have carried out this review. 
As far as, I am concerned, it is a very interesting review about glial cells.
Next, I am will make comments
to give a constructive criticism:
- Regarding the writing: the line 107 has a lack space. In the lane 48, “neural stem cells” should be changed to “neuronal stem cells”.
- From my point of view, I miss in the general discussion something about different microglial morphologies. Next, I quote an article on different microglial morphologies. Lier J, Streit WJ, Bechmann I. Beyond Activation: Characterizing Microglial Functional Phenotypes. Cells. 2021 Aug 28;10(9):2236. doi: 10.3390/cells10092236. PMID: 34571885; PMCID: PMC8464670. Finally, I want to congratulate the authors of this work again for the quality of the review.

Reviewer 2 Report

In this review, the authors describe the role of heterogeneous glial cell populations present in the CNS in the physiological conditions and that are also actively participating in pathophysiology following ischemic brain injury. The manuscript is overall well-written, documented with numerous (284) and relevant references providing many open questions regarding heterogeneity of glial cell (sub)types and sometimes contradictory roles in their response to injury. There are however few major points/issues that must be addressed by the authors:

-        There are 6 sections describing different glial cell types and their origin, but there is only one table (Table 1) that compares NG2 glia between mice and humans. It is not clear why for instance astrocytes -for which the classification is far from being definitive – are not described with accompanying figure/scheme/table showing different subtypes, morphologies and/or molecular/functional properties.  

-        Similarly, section 4 describes oligodendrocyte’s heterogeneity which could be nicely summarized with table reporting parameters considered in each subsection (classification, IHC markers, transcription factors etc). This would enhance the contribution of this review for better understanding of the current knowledge in the field, which is the main purpose of review articles. In the same time it will help the reader to sum up the main conclusions.

-        Section 2: Since it is increasingly recognized that NG2 are the third macroglia cell type, it should be important to explain in the same way to which glia category ependymal cells belong (some of the recent textbooks include them in astrocytes https://www.researchgate.net/publication/265601251_Introduction_to_Neuroglia) while some older one are showing them as separate category. In this work they are listed into astrocytes subtypes (lines 179-180). It would be helpful to extract this information from the references used to describe them.

-        The abbreviations are not consistently used: often there are repetition of already provided abbreviated forms: for instance, CNS is defined in the beginning (line 109) but it is repeated many times (lines 143, 184, 382 etc) and this is the case for many other abbreviations such as SVZ, OL, OPC, NSC, DG, BBB…  

-        Line 191: unnecessary repeating sentence: it was already said that astrocytes are components of BBB (line 133) while it is not clear if perivascular and marginal astrocytes (line 177-178) are distinct population respect to protoplasmic astrocytes that are also in contact with brain blood vessels. The repetitive sentences/parts should be avoided.

-        The claim that OPCs are “genetically” homogeneous cell population (line 494) is in net contradiction with what was stated in line 364 (ref. no. 82) and line 452. Since in the next sentence this is further explained, perhaps this sentence should be rephrased to avoid confusion, such as: “one could conclude that OPC represent a homogeneous cell population”

Minor points:

Abstract

Line 16: “known” is not necessary, “proliferative potential” alone explains their contribution in response to injury

Line 18: sentence requires rephrasing: “…neurogenesis, proliferation, and migration of neurons…” – as it is written one could conclude that neurons can proliferate (!)

Main text

Line 39: “as a scientific field” – unnecessary

Line 47: correct the part “adult rodents and mammals” - rodents are mammals!

Line 198-199: “electric” instead of “electronic”; the sentence is unclear

Line 253: correct the part “receptor in extracellur matrix” with “receptor for extracellur matrix components”

Line 298: “wild type” is commonly abbreviated into “wt”; using “wt mice” instead of “(wild type)” facilitates the reading (same for line 577)

Line 321: “their progression”: it is not clear the meaning- “development” is more precise term

Line 317: correct the following part of the sentence: “non-human primates, including humans”

Lines 346-349: the sentences could be combined since in the first one it is not clarly explained the comparison: “larger/longer” between human protoplasmic astrocytes and other species? Or vs other astrocytes' types?

Line 370: “regional loss of gene expression” – of which genes? Sentence needs rephrasing.

Line 373-374: if “genetic defect” refers to “knockout” that is enough to define it immediately

Line 406-407: Myelin as well as all other cellular structures for which specific antibodies are raised have antigenic properties. I suggest to remove this sentence.

Line 419-420: “only at certain moments” – “transiently” is more specific term

Lines 420, 432; “cell(ular) progression” requires more precise definition (toward which process-proliferation, differentiation, maturation?)

Line 492: “region chosen for proliferation” could (should) be rephrased

Line 507-508: consider to use “differentiate” instead of “transform” since the cell transformation is often used in field of cancer research or, alternatively is used to define incorporation of exogenous genetic materials (e.g. cell transfections)

Line 574: “gene markers” could be confusing – differential expression of specific genes can be used as markers (while microglia genome is the same)

Lines 665-667: check the sentence: not all protein expression profiles lead to morphological changes (as it is written). In the next sentence it should be also clear that “the extent of observed changes” depends on the severity of the injury.

Line 726: explain better the phenomenon of mitochondria transfer (from-to which cells?)

Line 735: experiments on animals

Line 749: “unlimited plasticity of reactive glia” – overstated

Line 796: “significance” – role or function is perhaps more appropriate term

Line 882: reference is missing

Line 933: since Sun et al paper is cited and in that paper authors propose SOX9 as reliable marker of astrocytes in adult mouse brain (except for the neurogenic niches), it should be more clearly explained that even if it “regulates the processes of differentiation in both astrocytes and oligodendrocytes” its expression is astrocyte-specific.

Table 1

Line 534: it is not clear if only reference no. 30 was used to create Table 1 or other references cited in the text were used as well.

In the table (the column “Parameters”) it would be better to put:

Gray matter localization

White matter localization

Gap junctions (remove “between them”)– Yes or no

The English language requires minor editing.
